# Unleashing the Representational Power of Fourier Shapes for Attacking Infrared Object Detection

**Yixing Yong** [1]  **Jian Wang** [1]  **Ming Lei** [2]  **Lijun He** [1]  **Fan Li** [1 3]

## Abstract

Infrared object detection is crucial for perception in autonomous driving and surveillance but remains vulnerable to physical adversarial attacks. Unlike in the RGB domain, where attacks rely on color texture, infrared attacks must manipulate thermal signatures, making the **geometry shape** of heat-blocking materials the primary adversarial information carrier. Current shape-based methods suffer from a fundamental trade-off between **representational capability** and **optimization power**, limiting their attack effectiveness. In this work, we overcome this dilemma by introducing learnable Fourier shapes to the infrared domain. We utilize an end-to-end differentiable framework where a compact set of Fourier coefficients, defining the shape boundary, is analytically mapped to a pixel-space mask via the winding number theorem. This enables efficient gradient-based optimization to generate potent shapes that cause human targets to evade detection. Extensive digital and physical experiments provide a comprehensive evaluation and validate our superior performance. Our resulting physical patch achieves striking robustness, successfully evading detectors across diverse distances, angles, poses, and individuals, and achieves over $88\%$ attack success rate at distances greater than 25m (conf.=0.5). Code is available at https://github.com/Yongyx99/Fourier-shape-attack.

[1]School of Information and Communications Engineering, Faculty of Electronic and Information Engineering, Xi'an Jiaotong University, Xi'an, China [2]School of Physics, Xi'an Jiaotong University, Xi'an, China [3]School of Computer Science and Technology, Xinjiang University, Urumqi, China. Correspondence to: Fan Li <lifan@mail.xjtu.edu.cn>.

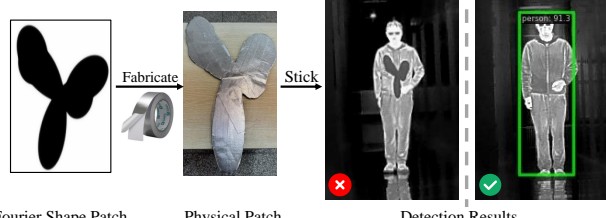

*Figure 1.* Physical Fourier shape attack against infrared detection. We optimize a Fourier shape, fabricate it from heat-blocking material, and apply it to a person. This adversarial shape renders the person invisible to the infrared detector, while the benign person is easily detected.

## 1. Introduction

Deep Neural Networks (DNNs) are widely used for environmental perception, demonstrating exceptional performance in various downstream tasks such as object detection (Wang et al., 2024; Zhao et al., 2025; Chen et al., 2025), segmentation (Kirillov et al., 2023; Yang et al., 2024), and recognition (Yang et al., 2021; 2020). However, a widely recognized drawback is their vulnerability to adversarial examples (Wang et al., 2023; Zhou et al., 2024; Yuan et al., 2025), which are inputs meticulously designed to induce any desired incorrect outputs. This vulnerability extends beyond digital-only manipulations and poses a direct threat in the physical world (Zheng et al., 2024; Wei et al., 2024a). For instance, in the domain of natural visible light (RGB) images, adversarial patches (Cheng et al., 2024; Hu et al., 2023) have been highly effective. These printed patterns, when applied to objects, can successfully fool detectors, causing missed detections of pedestrians or vehicles.

This attack paradigm, however, encounters a fundamental obstacle when applied to different imaging modalities. Thermal infrared cameras (Bustos et al., 2023; Zhao et al., 2024), valued for their robustness to variable lighting and their widespread use in security, surveillance, and autonomous driving, operate on a different imaging mechanism. They capture thermal radiation, not reflected light, rendering color-based patches ineffective. To attack these systems, one must manipulate the object's perceived thermal signature (Wei et al., 2024b; 2023c; Zhu et al., 2021). This has

led to the exploration of heat-blocking materials, such as aerogel or aluminum film. A patch cut from such a material and placed on a person, for example, creates a distinct cold region (Wei et al., 2023a) in the thermal image. In this context, **the geometry or shape of the patch, rather than its color, becomes the carrier of adversarial information.**

Despite this novel approach, the performance of current shape-based infrared attacks (Wei et al., 2024b; 2023c; Zhu et al., 2021; 2022) remains significantly weaker than that of pixel-optimized patches designed for RGB images. While existing works often claim high attack success rates (ASR) (Wei et al., 2024b; 2023c; Zhu et al., 2024), these results frequently rely on permissive evaluation standards. For example, success is often declared if a target's detection confidence merely drops below a high threshold like 0.7. We find this masks a crucial weakness: these methods only achieve a minor suppression of detector confidence. By slightly lowering the detection confidence threshold, the ASR can drop dramatically; for example, reducing the threshold from 0.7 to 0.4 can cause the ASR to collapse from over 90% to below 20%, revealing their inability to robustly conceal a target.

This poor performance originates from a fundamental compromise between **representational capability** and **optimization power** in current shape-generation techniques. Some methods prioritize differentiability by modeling the shape as a discrete 2D grid (Wei et al., 2024b; 2023d;a), treating each pixel as an optimizable parameter. This bottom-up approach, while compatible with gradient descent, provides a crude approximation of shape and must rely on complex auxiliary losses, like pixel aggregation (Wei et al., 2023d), just to ensure the final output is a cohesive, physically plausible patch. Conversely, other methods seek representational capability using spline-interpolated vertices to define a smooth boundary (Wei et al., 2023b;c). This, however, forfeits optimization power, as the lack of a differentiable pipeline necessitates the use of inefficient, black-box search algorithms. These algorithms struggle in the high-dimensional search space and rarely find optimal solutions.

We overcome this dilemma by unifying a powerful parametric shape representation with an efficient, end-to-end differentiable optimization framework. This work, for the first time, adapts learnable Fourier shapes to the domain of adversarial infrared detection. We model the physical heat-blocking patch as a 2D shape whose boundary is defined by a compact set of Fourier series coefficients. This abstract representation is then analytically rasterized onto a 2D grid using a fully differentiable mapping module based on the winding number theorem. This module creates a seamless, differentiable bridge to the DNN, allowing the adversarial loss to be backpropagated directly to the Fourier coefficients.

This enables us to learn a highly effective adversarial shape using standard gradient descent.

**Our framework offers several distinct advantages over prior art. First**, it employs a top-down, holistic shape definition. A Fourier series inherently describes a complete, closed contour, thus completely obviating the need for the complex pixel-level aggregation constraints required by grid-based methods. **Second**, the Fourier representation offers unparalleled expressive power and flexibility. By simply increasing the number of Fourier terms, our model can represent arbitrarily complex geometries, granting it access to a vast and rich search space with minimal impact on computational complexity. **Third**, and most critically, the differentiable mapping module provides a robust analytic bridge between the shape parameters and the pixel space, enabling the use of powerful, gradient-based optimization and ensuring we find potent adversarial solutions.

We conduct extensive experiments in both digital and physical settings, demonstrating that our Fourier-based adversarial shapes significantly outperforms existing methods in concealing human targets from infrared detectors. The example of our proposed infrared adversarial patch is shown in Figure 1, and the comparison with existing infrared adversarial attacks is shown in Figure 2. Furthermore, recognizing the inconsistent evaluation protocols in prior work, we propose a rigorous and standardized set of metrics to serve as a benchmark for future research in this area. Our contributions can be summarized as follows:

- We analyze the shortcomings of existing infrared adversarial attacks, identifying the conflict between shape representation and optimization power as the key performance bottleneck.

- We are the first to introduce learnable Fourier shapes to the infrared domain, creating an end-to-end differentiable framework for generating potent physical adversarial patches.

- We conduct extensive experiments to validate the superior performance of our method and establish a much-needed benchmark for rigorously evaluating physical attacks in the infrared domain.

## 2. Related Work

Research on adversarial patches has primarily focused on optimizing the pixel content within a fixed, regular shape (e.g., a square) (Thys et al., 2019; Guesmi et al., 2024). These methods, whether based on direct pixel optimization (Wang et al., 2025b;c; Hu et al., 2022), Generative Adversarial Networks (GANs) (Yong et al., 2025; Hu et al., 2021; Wang et al., 2025a), or diffusion models (Wang et al., 2025e; Wei et al., 2025), generate textures that effectively deceive

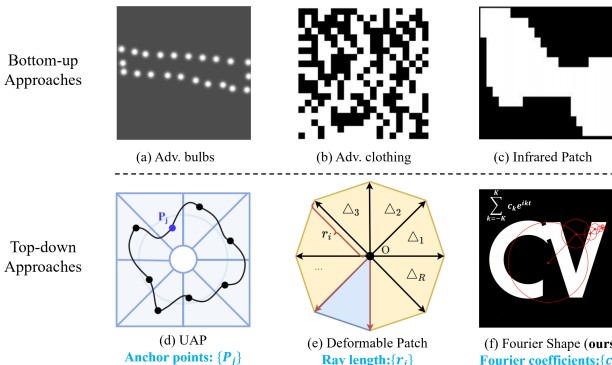

Bottom-up Approaches

(a) Adv. bulbs    (b) Adv. clothing    (c) Infrared Patch

Top-down Approaches

(d) UAP
**Anchor points: $\{P_j\}$**

(e) Deformable Patch
**Ray length:$\{r_i\}$**

(f) Fourier Shape (**ours**)
**Fourier coefficients:$\{c_k\}$**

*Figure 2.* The comparison between different shape-based infrared attacks. Blue denotes the corresponding shape definition parameters in top-down approaches. (a)-(e) represent ref. (Zhu et al., 2021), (Zhu et al., 2022), (Wei et al., 2023d), (Wei et al., 2023b), and (Chen et al., 2022), respectively.

object detectors for tasks like person or vehicle detection. However, this pixel-domain optimization often results in high-frequency, unnatural patterns, which are conspicuous to human observers and easily defended.

A separate line of inquiry explores shape-based attacks, which optimize an object's geometry while keeping its texture fixed. These methods fall into two categories. Bottom-up approaches (Wei et al., 2024b; 2023d;a; Zhu et al., 2022; 2021) define the shape on a discrete pixel grid. While compatible with gradient-based methods, this representation is crude and requires complex auxiliary losses, such as aggregation penalties (Wei et al., 2024b; 2023d), to enforce geometric cohesion. Top-down approaches offer superior representation. For instance, Wei *et al.* (Wei et al., 2023b;c) use spline-interpolated vertices to model smooth contours. However, this pipeline is non-differentiable, forcing reliance on inefficient black-box optimization (e.g., Differential Evolution). Chen *et al.* (Chen et al., 2022) bridge this gap by proposing a differentiable, ray-based representation, but its modeling capability is strictly limited to star-shaped polygons and cannot express arbitrary contours.

In summary, existing methods are forced to trade off between shape representational capability and optimization power. Following Wang *et al.* (Wang et al., 2025d), which used Fourier series to represent arbitrary 2D closed curves, we are the first to introduce this concept to infrared adversarial attacks and validate its feasibility in the physical world.

## 3. Method

We adopt an end-to-end differentiable pipeline for generating physically realizable, shape-based adversarial attacks targeting infrared object detectors, as shown in Figure 3. The core idea is to optimize the geometric boundary of a physical heat-blocking patch, represented by a compact

set of Fourier coefficients. This section details our attack pipeline, the parametric shape representation, the differentiable mapping process, and the optimization objectives designed to create potent and plausible adversarial shapes.

### 3.1. Problem Definition and Attack Pipeline

Let $I \in \mathbb{R}^{H \times W}$ be a clean, infrared image, where pixel values correspond to thermal intensity. Let $B = (x, y, w, h)$ be the ground-truth bounding box for a target object, such as a pedestrian, within $I$. We denote the infrared object detector as $f(\cdot)$. Given an input image, the detector $f$ outputs a set of raw proposals $P = \{p_i\}$, where each proposal $p_i = (b_i, s_i)$ consists of a predicted bounding box $b_i$ and an objectness confidence score $s_i$.

Our goal is to generate a shape, represented by a continuous mask $M_s \in [0, 1]^{H_s \times W_s}$. This shape mask $M_s$ is generated from a set of optimizable Fourier coefficients $\Theta$ via a function $G$, such that $M_s = G(\Theta)$ (detailed in Sec. 3.2 and 3.3). When applied to the target, this mask is intended to maximally suppress the confidence scores $s_i$ for all proposals $p_i$ associated with $B$. The values in $M_s$ define the shape, where a value approaching 1 signifies a region to be covered by heat-blocking material (which appears *cold* or *black* in the infrared image), and a value of 0 represents the original, unobstructed thermal signature.

To perform the attack, we define the transformation $\mathcal{T}(M_s, B, \rho)$ that scales the $M_s$ based on the target's bounding box $B$ and a scale ratio $\rho$. This ratio $\rho$ controls the patch size relative to the target's bounding box; for example, a value of $\rho = 1$ allows the patch to span the full width and height of $B$. The function then translates the scaled mask to the center of $B$. This process yields a full-image mask $M_{applied}$ with the same dimensions as $I$. While $B$ is defined here as a single box, this formulation applies to multiple targets without loss of generality.

The adversarial image $I_{adv}$ is then generated by applying the shape patch onto the original image. Assuming the heat-blocking area has a pixel value of 0, this process is formulated as:

$$I_{adv} = I \odot (1 - \mathcal{T}(M_s, B, \rho)) \tag{1}$$

where $\odot$ denotes element-wise multiplication.

Our end-to-end pipeline optimizes the shape parameters $\Theta$ by minimizing a loss function $\mathcal{L}$ computed from the detector's output on the adversarial image. Once optimization converges, the final mask $M_s$ defines the precise shape to be physically cut from a heat-blocking material and deployed for a real-world attack.

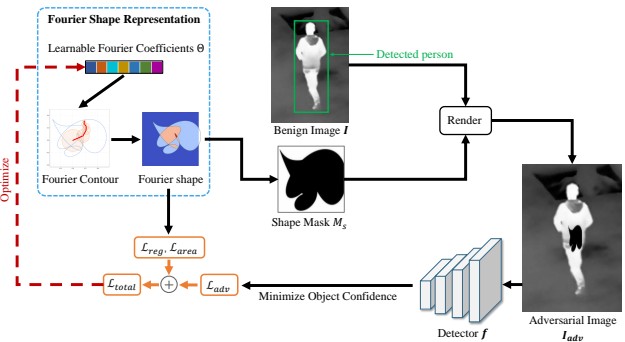

*Figure 3.* The overall framework of the Fourier shape attack.

## 3.2. Fourier Shape Representation

To overcome the limitations of existing shape models, we require a representation that is both highly expressive and inherently describes a complete, physically plausible contour. Grid-based methods are difficult to constrain, while spline-based methods lack efficient optimization. We therefore adopt a powerful parametric model, defining the 2D shape boundary as a Fourier series in the complex plane.

A shape boundary $F(t)$ is represented as a function of a parameter $t$ that traverses the contour from 0 to $2\pi$:

$$F(t) = f(t) + i \cdot g(t) = \sum_{k=-K}^{K} c_k e^{ikt} \quad \text{for} \quad t \in [0, 2\pi] \tag{2}$$

Here, $f(t)$ and $g(t)$ are the Cartesian coordinates of the boundary, $i$ is the imaginary unit, and $K$ is the highest frequency component, which dictates the shape's complexity. The shape is entirely defined by the set of complex Fourier coefficients $\Theta = \{c_k\}_{k=-K}^{K}$. These coefficients are the compact and continuous set of parameters we optimize.

This top-down representation is particularly advantageous for our task. Any set of coefficients $\Theta$ inherently defines a single, closed contour, which guarantees physical realizability without the complex aggregation losses required by discrete grid-based methods. The coefficients also have intuitive geometric roles: $c_0$ defines the shape's center, $c_1$ and $c_{-1}$ establish its fundamental elliptical form (scale and orientation), while higher-order harmonics ($|k| \geq 2$) contribute progressively finer details and complexity.

## 3.3. Differentiable Shape-to-Image Mapping

A key challenge is to create a differentiable bridge from the abstract Fourier coefficients $\Theta$ to the discrete pixel mask $M_s$ that our attack pipeline requires. We achieve this using a differentiable calculation based on the winding number theorem, which analytically connects the continuous boundary $F(t)$ to a 2D pixel grid.

The winding number $W(q)$ quantifies how many times the curve $F(t)$ travels counter-clockwise around a given point $q = (x_q, y_q)$. For a simple, non-self-intersecting curve, $W(q)$ is 1 for any point $q$ inside the curve and 0 for any point $q$ outside. This property provides a perfect criterion for defining the shape's interior. The winding number is computed by the line integral:

$$W(q) = \frac{1}{2\pi} \int_0^{2\pi} \frac{(f(t) - x_q)g'(t) - (g(t) - y_q)f'(t)}{(f(t) - x_q)^2 + (g(t) - y_q)^2} dt \tag{3}$$

In our implementation, we evaluate this integral for every pixel coordinate $q$ in our $H_s \times W_s$ mask grid. The integral is approximated as a differentiable sum over $N_s$ discrete sample points $t_j$ along the curve. This process yields a raw winding number image $I_W$, where each pixel's value $I_W(q)$ is a continuous, floating-point approximation of $W(q)$.

During optimization, the shape may self-intersect, leading to values in $I_W$ that can be positive, negative, or have magnitudes greater than 1. To convert this raw output into our final patch mask $M_s \in [0, 1]$, we first take the absolute value of the raw winding number. This step ensures that all interior regions (which have non-zero winding numbers) are treated as part of the patch. We then clip the result to the range $[0, 1]$ to finalize the mask. This normalization effectively thresholds the continuous field, mapping all significant interior regions to 1 and the exterior to 0. This two-step process is combined as:

$$M_s(q) = \text{Clip}(|I_W(q)|, \max = 1) \tag{4}$$

This entire mapping process $G : \Theta \to M_s$ is fully differentiable, which is the key component that allows gradients to flow from the optimization objective back to the shape defining parameters $\Theta$.

## 3.4. Optimization Objectives

Our final optimization goal is to find the shape $\Theta$ that minimizes a composite loss function $\mathcal{L}_{total}$. This loss is a weighted sum of three components: an attack loss $\mathcal{L}_{adv}$, an area constraint $\mathcal{L}_{area}$, and a shape regularization $\mathcal{L}_{reg}$.

$$\mathcal{L}_{total} = \mathcal{L}_{adv} + \alpha\mathcal{L}_{area} + \beta\mathcal{L}_{reg} \tag{5}$$

where $\alpha$ and $\beta$ are hyperparameters balancing the contribution of each term.

**Attack Loss.** To ensure the target is consistently concealed, we must suppress all detection proposals associated with it, not just the one with the highest score. Therefore, we compute the adversarial loss on the set of all proposals $\hat{P} \subseteq P$ associated with $B$ by minimizing the confidence scores:

$$\mathcal{L}_{adv} = -\sum_{p_i \in \hat{P}} \log(1 - s_i) \tag{6}$$

Note that $P$ is the raw output from the detector before Non-Maximum Suppression (NMS) is applied.

**Area Constraint.** For a physical attack, the adversarial patch should be as small as possible to improve stealthiness and ease of deployment. We directly enforce this by minimizing the mean value of the patch mask $M_s$. This area loss is defined as:

$$\mathcal{L}_{area} = \frac{1}{H_s \times W_s} \sum_{q \in M_s} M_s(q) \qquad (7)$$

This encourages the optimizer to find the smallest possible shape that still achieves a successful attack.

**Shape Regularization.** Unconstrained optimization can produce shapes with excessive high-frequency noise, resulting in *jagged* or *spiky* boundaries. Such shapes are difficult to cut accurately and are less physically plausible. To promote smooth, realistic shapes, we enforce a constraint on the Fourier coefficients. We require the shape's structure to be dominated by its low-frequency components. This is achieved by penalizing any high-frequency harmonic amplitude $|c_k|$ (for $|k| \geq 2$) that exceeds a fraction $\gamma$ of the total fundamental amplitude $S_{fund} = |c_1| + |c_{-1}|$. This regularization loss is formulated as:

$$\mathcal{L}_{reg} = \sum_{|k|=2}^{K} \text{ReLU}(|c_k| - \gamma S_{fund}) \qquad (8)$$

This loss term acts as a prior for smooth geometries, guiding the optimization towards plausible shapes that are effective in simulation and practical for physical deployment.

**The theoretical analysis of the Fourier Series' ability to infinitely represent 2D closed shapes** can be found in the Appendix.

# 4. Experiments

## 4.1. Experimental Setup

**Dataset:** For our digital attack experiments, we utilize the infrared modality images from the LLVIP dataset (Jia et al., 2021), focusing on the pedestrian class. Following (Wei et al., 2023b;c), we select around 120 pedestrian instances from the test set on which the detector achieves the highest confidence scores to serve as our attack set. Consequently, the clean Average Precision (AP) on this specific subset is 100%. All detectors are trained on the official training set.

**Target detector:** We choose the widely-used YOLOv3 (Redmon & Farhadi, 2018) as the baseline detector for comparing against SOTA attack methods. Additionally, we conduct experiments on Faster R-CNN (Ren et al., 2016),

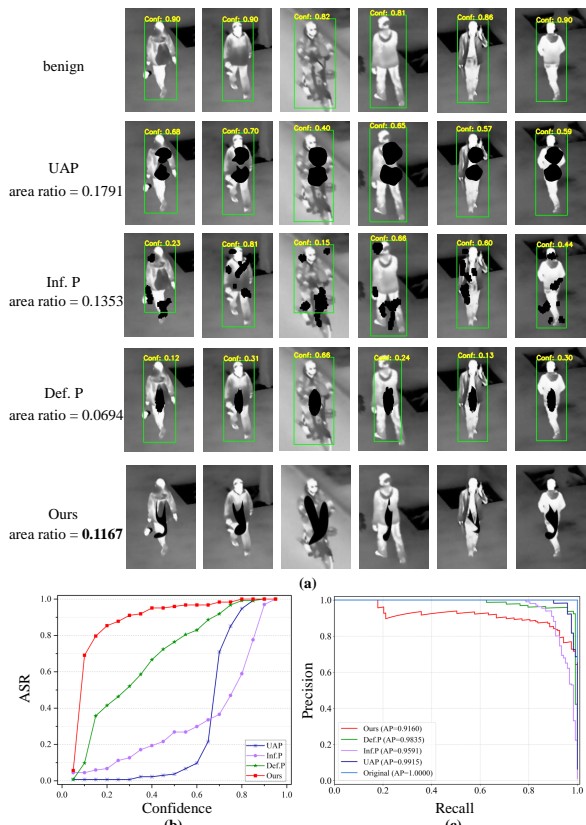

*Figure 4.* Comparison results with previous attacks on YOLOv3. (a) Visualizations of attack results. Area ratio is used to reflect the size of adversarial patches. The object confidence score here is set as 0.1. (b) ASR-Confidence curve. (c) Precision-Recall curve.

RetinaNet (Lin et al., 2017) and modern YOLOv8 (Jocher et al., 2023) to verify the generalizability of our approach.

**Evaluation Metrics:** We evaluate attack performance using ASR and AP drop. However, we depart from the common practice of evaluating ASR at a single, fixed confidence threshold, such as ASR@0.5 or ASR@0.7. We argue that this approach is insufficient because detection thresholds are flexibly adjusted in real-world scenarios, where they are often lowered to avoid missed detections (false negatives) or raised to reduce false positives. The high ASR reported by existing methods at a high threshold is often fragile, as their effectiveness may collapse dramatically with only a slight reduction in this threshold. Therefore, to provide a more comprehensive and practical assessment, we analyze the ASR across a full range of confidence thresholds by presenting a complete ASR-threshold curve.

**Implementation:** The Fourier coefficients $\Theta$ is optimized using the Adam for up to 1000 iterations with a learning rate of 0.002. By default, the highest Fourier frequency is set to $K = 6$, the scale ratio $\rho$ (from Eq. 1) is 0.6, and the loss weighting coefficients are set to $\alpha = 1$ and $\beta = 0.1$.

See Appendix for more training details.

## 4.2. Comparisons with SOTA Methods

We select three SOTA shape-based attacks for comparison. Specifically, UAP (Wei et al., 2023b) uses spline-interpolated vertices to define a shape with a smooth boundary and optimizes anchor point positions using a black-box evolution. Inf. P (Wei et al., 2023d) models the shape using a discrete pixel grid and employs an aggregation constraint to ensure geometric cohesion. Def. P (Chen et al., 2022) adopts a ray-based shape representation. When reproducing these methods, we adopt maximum area definition parameters from the original papers to generate the shapes. Details see Appendix. Figure 4 presents a comprehensive comparison on the YOLOv3 detector.

**Qualitative Analysis.** The visualization results are shown in Figure 4(a). To intuitively illustrate the patches' effects on the detector, we visualize all detection results with a confidence score above 0.1 (a very low threshold). For UAP, the attack is largely ineffective, a result of its inefficient black-box optimization, which only manages to slightly reduce the target's confidence. While Inf. P employs an aggregation constraint, the resulting shape is visibly dispersed and fragmented to achieve its attack, posing significant challenges for deployment as a single, cohesive physical patch. For Def. P, the adversarial information is heavily concentrated in the shape's high-frequency edges. This makes the attack fragile and highly sensitive to physical-world inaccuracies, such as cutting errors, which can lead to a loss of effectiveness. In contrast, our method, powered by the superior representational capacity of Fourier series, generates potent and cohesive shapes. As shown, our attack successfully suppresses all target confidences to below the 0.1 threshold, effectively hiding the pedestrians from the detector.

**Quantitative Analysis.** The ASR-Confidence curve is shown in Figure 4(b), where a higher threshold indicates a more lenient evaluation for the attack. The performance of both UAP and Inf. P is revealed to be highly sensitive to this threshold. For instance, while UAP achieves an ASR of approximately 95% at a lenient 0.8 threshold, its ASR collapses to just 10% when the threshold is lowered to 0.6. This fragility confirms that while prior works may report high ASR, this success is often superficial and not robust. Our method consistently and significantly outperforms all SOTA methods across all confidence thresholds, achieving a high ASR even at very stringent (low) thresholds.

In addition, Figure 4(c) shows the Precision-Recall (P-R) curves, where the Area Under the Curve (AUC) is the AP. While our method achieves the largest AP drop (from 1.0 to 0.9104), the AP reduction for all methods appears less dramatic than often reported. This is for two reasons. First, we compute AP using the *full* P-R curve, unlike prior works

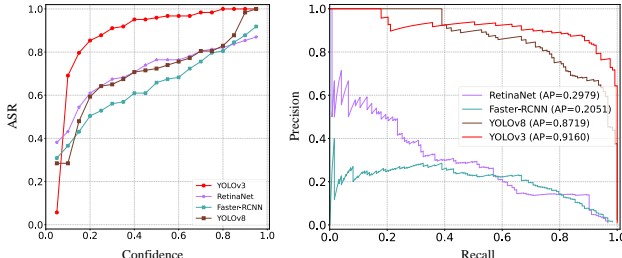

*Figure 5.* Attack results on different detectors. (a) ASR-Confidence curve; (b) P-R curve.

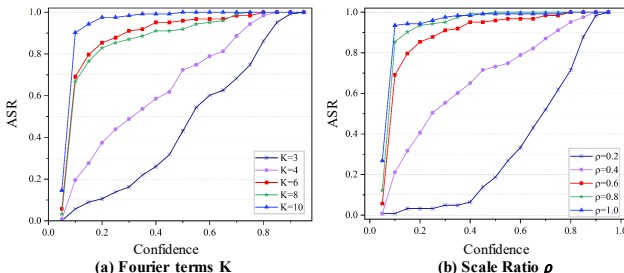

*Figure 6.* Ablations studies. The ASRs on YOLOv3 are presented. (a) Fourier terms $K$; (b) Scale ratio $\rho$.

that often use a *truncated* curve (e.g., $\geq 0.5$ confidence), which does not account for the mass of low-confidence proposals and may lead to an overestimation of the perceived AP drop. Second, as shown in visualization results, the attack's primary effect is suppressing confidence scores ($s_i$) rather than altering box locations ($b_i$). If an attack merely lowered all confidences uniformly without changing their locations, the AP drop would be zero. This highlights that AP drop and the ASR-Confidence curve are complementary, and both are needed to fully understand an attack's impact.

**Area Efficiency.** Since these methods lack explicit parameters to strictly fix the final patch size, we evaluate the mean area ratio, defined as the patch area relative to the target bounding box, after optimization. As shown in Figure 4, our method achieves the most potent suppression while maintaining a highly competitive area ratio of 0.1167.

## 4.3. Attack on Different Detectors

To validate the generalizability of Fourier-based shape attack, we extend our evaluation to other mainstream detectors: RetinaNet (Lin et al., 2017), Faster R-CNN (Ren et al., 2016) and YOLOv8 (Jocher et al., 2023). For the two-stage Faster R-CNN, the adversarial loss is calculated to simultaneously suppress detection outputs from both the Region Proposal Network (RPN) and the Region of Interest (RoI) refinement stage.

The experimental results are presented in Figure 5. The ASR-Confidence curve demonstrates potent and robust ef-

Table 1. Quantitative ablations on the the number of Fourier terms $K$ and the scale ratio $\rho$.

| Effect of $K$ ($\rho = 0.6$) | | | | | |
|---|---|---|---|---|---|
| $K$ | **3** | **4** | **6** | **8** | **10** |
| Conf. drop ↑ | 0.3245 | 0.5054 | 0.7178 | 0.7043 | 0.7762 |
| AP drop ↑ | 0.0031 | 0.0246 | 0.0840 | 0.1066 | 0.1132 |

| Effect of $\rho$ ($K = 6$) | | | | | |
|---|---|---|---|---|---|
| $\rho$ | **0.2** | **0.4** | **0.6** | **0.8** | **1.0** |
| Conf. drop ↑ | 0.2029 | 0.5321 | 0.7178 | 0.7646 | 0.7786 |
| AP drop ↑ | 0.0035 | 0.0268 | 0.0840 | 0.2053 | 0.2473 |

Table 2. Quantitative ASR of the physical attack as a person walks continuously from 45m to 15m.

| Distance Range | ASR@0.1 | ASR@0.3 | ASR@0.5 |
|---|---|---|---|
| 35m–45m | 10% | 92% | 100% |
| 25m–35m | 11% | 60% | 88% |
| 15m–25m | 4% | 16% | 31% |

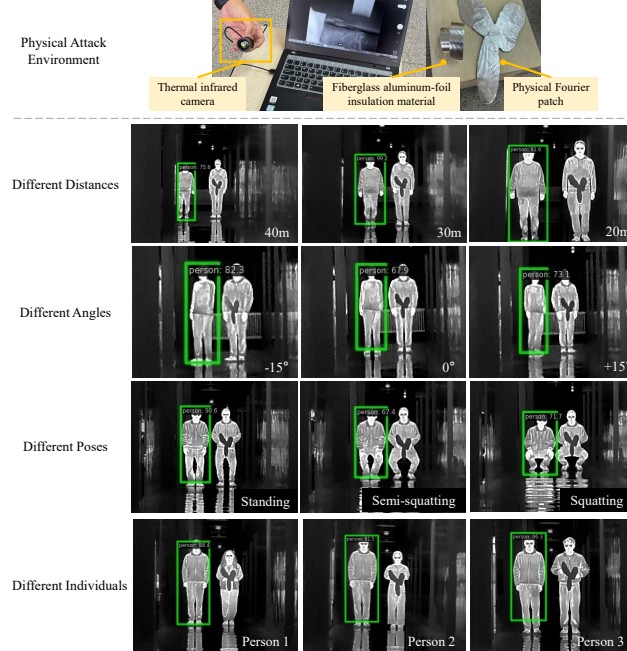

Figure 7. The physical attack environment and the attack results across different distances, viewing angles, body poses, and individuals.

fectiveness against all four architectures. The attacks consistently achieve high ASR values and are not sensitive to the confidence threshold, confirming that our approach is not tailored to a single detector. Furthermore, from the P-R curve we observe that while the AP drop on YOLOv3 and YOLOv8 were modest, the attack induce a catastrophic performance collapse in the other two detectors. The AP for RetinaNet drops from 1.0 to 0.2979, and for Faster R-CNN, it collapses to 0.2051. This strongly suggests that the modest AP drop is a behavior specific to the YOLO models, not a weakness of our attack method. Additionally, attack results on YOLOv8 indicate that our method is effective not only against conventional detectors, but also extends to more modern detectors.

### 4.4. Ablation Study

We ablate the number of Fourier terms $K$ (shape complexity) and the scale ratio $\rho$ (patch size). The ASR-Confidence curves are shown in Figure 6, and detailed quantitative results, including *mean confidence drop* and *AP drop*, are provided in Table 1.

**Fourier terms** $K$. As shown in Figure 6(a), a higher $K$ increases shape complexity and generally improves attack performance. However, this performance gain saturates beyond $K = 6$. Considering that higher $K$ values also increase optimization difficulty and create shapes that are harder to physically fabricate, we select $K = 6$ as the best compromise between effectiveness and physical plausibility.

**Scale Ratio** $\rho$ controls the patch area relative to the target's bounding box. As shown in Figure 6(b), attack strength generally increases with $\rho$, as a larger patch is more effective but also more conspicuous. Therefore, we set $\rho = 0.6$ by default.

### 4.5. Physical Attack

To validate the real-world efficacy of our method, we translate the digitally optimized shape into a physical attack. The

adversarial Fourier shape is first generated in the digital domain against the YOLOv3 detector. We then fabricate the physical patch by precisely cutting the optimized geometry from a sheet of *fiberglass aluminum-foil insulation material* (see Figure 7).

The qualitative results are presented in Figure 7. In each test scenario, a control person (without a patch) stands next to an attacker holding the physical Fourier patch. The detector consistently identifies the control person with high confidence. In contrast, the person holding our patch successfully evades detection in all trials. We demonstrate that this attack is highly robust and effective across a comprehensive range of real-world variations, including different distances (20m to 40m), viewing angles (-15° to +15°), body poses (e.g., standing, semi-squatting), and different individuals.

This robust physical performance is particularly noteworthy. Although the patch is optimized in an input-specific manner (trained on a single image), a process which typically struggles with generalization, it demonstrates remarkable

*Table 3.* Quantitative ASR of the physical attacks under different conditions.

| Environments | Settings | ASR@0.3 | ASR@0.5 | ASR@0.7 |
|---|---|---|---|---|
| | Camera shake | 11% | 48% | 78% |
| | Subject moving | 12% | 74% | 98% |
| Daytime (18°C) | Camera moving | 24% | 44% | 80% |
| | Different clothing | 25% | 98% | 100% |
| | Rear-view placement | 48% | 100% | 100% |
| Nighttime(10°C) | Person standing still | 23% | 96% | 100% |

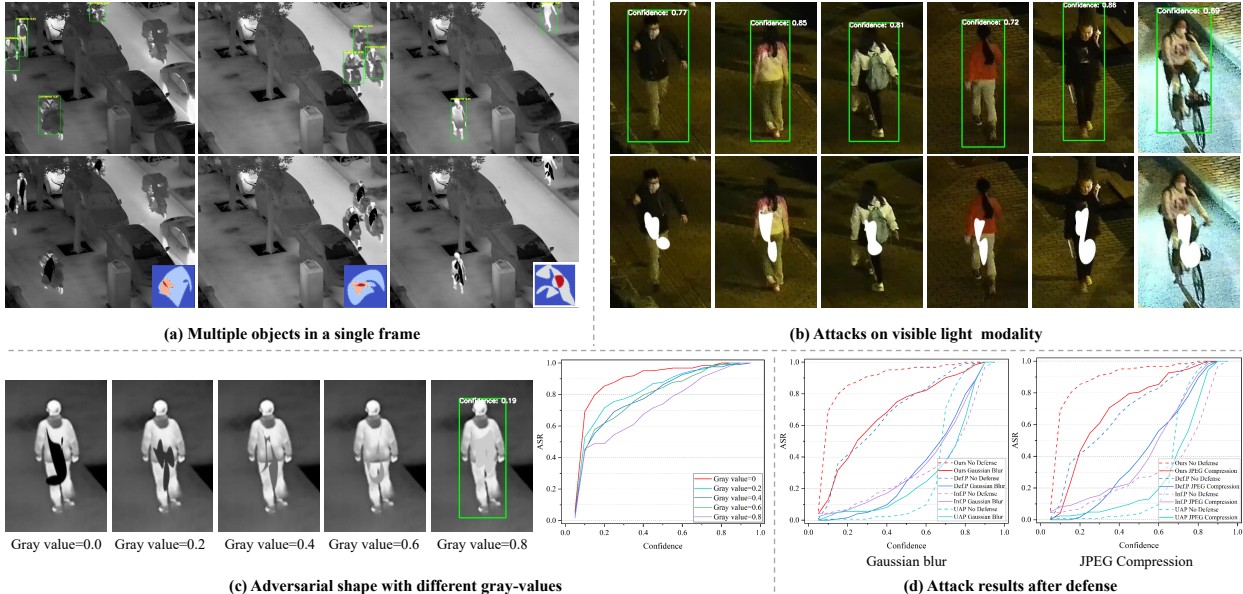

**(a) Multiple objects in a single frame**

**(b) Attacks on visible light modality**

Gray value=0.0   Gray value=0.2   Gray value=0.4   Gray value=0.6   Gray value=0.8

**(c) Adversarial shape with different gray-values**

Gaussian blur    JPEG Compression

**(d) Attack results after defense**

*Figure 8.* Analysis of the attack's generalizability and robustness. (a) Attacking multiple objects. (b) Attacking the visible light modality. (c) ASR sensitivity to patch gray-values. (d) Robustness against defenses. All results are against YOLOv3. Visualizations (a-c) use a 0.1 confidence threshold.

real-world transferability. This robustness is by design: we incorporate a set of data augmentations during the digital optimization phase, including random scaling, minor aspect ratio perturbations, and additive noise. This strategy forces the optimizer to learn a shape inherently robust to the inevitable sim-to-real gap, confirming the practical viability of our approach.

We further provide a quantitative evaluation by having a person walk continuously from 45m to 15m, calculating ASR over three distance intervals. The results, shown in Table 2, confirm that the attack is highly effective at medium-to-long distances (25m–45m), but its performance degrades as the target moves closer (15m–25m), where the detector captures stronger object details.

To evaluate the robustness of proposed adversarial attack, we conduct quantitative tests under diverse physical conditions, including object motion, camera shake/motion, clothing changes, rear-view placement, and diurnal time changes.

In each scenario, the shape patch is applied to a person 20 meters away from camera. The mean ASR is then evaluated across varying YOLOv3 detection confidence thresholds. Experimental results shown in Table 3 demonstrate that the proposed method exhibits adversarial effectiveness under various physical conditions, proving the robustness of proposed method.

### 4.6. Discussion

**Attacking Multiple Objects.** The aforementioned digital attack experiments are conducted on images containing only a single person per frame following (Wei et al., 2023b;c; 2024b; 2023d). However, real-world scenes often contain multiple targets. We investigate our framework's ability to find a single, universal shape for such scenarios. We apply the same optimizable Fourier shape to all person instances in the frame and jointly optimized the Fourier coefficients. As shown in Figure 8(a), optimized shape successfully fools the detector, causing all targets in the frame to evade detection.

**Attacks on Visible Light Modality.** To test the generality of Fourier shape beyond the infrared domain, we apply it to the visible light modality from the LLVIP dataset. We reconfigure the attack pipeline to render a solid white patch and re-optimize the Fourier coefficients from scratch against the visual detector. Figure 8(b) demonstrates that the attack is equally effective, proving that our differentiable shape optimization is a general, modality-agnostic framework.

**Sensitivity to Patch Gray-Values.** Ideal thermal attacks assume perfect heat-blocking (a gray-value of 0.0 in the patched area). In practice, materials may be imperfect. We simulate this by retraining the patch with different gray-values. As shown in Figure 8(c), the ASR degrades monotonically as the patch becomes less *cold* (i.e., the gray-value increases). Even at a gray-value of 0.8, the attack retains considerable effectiveness, even at lower confidence thresholds. This highlights that while thermal-blocking efficiency impacts performance, our method provides robustness against imperfect materials.

**Robustness Against Defenses.** Figure 8(d) demonstrates that our attack exhibits superior resistance to common defenses, including Gaussian blur and JPEG compression. Furthermore, the results of using adversarial augmentation as a defense are discussed in the Appendix.

## 5. Conclusion

We introduce an end-to-end differentiable framework for generating physical infrared attacks using learnable Fourier shapes. This approach resolves the critical trade-off between shape representation and optimization power. Extensive experiments establish a rigorous benchmark, demonstrate SOTA performance, and confirm the attack's real-world viability. Theoretically, the near-infinite representational capacity of Fourier shapes shifts the core adversarial challenge from shape representation to effective optimization.

## Acknowledgment

This work was supported in part by the National Natural Science Foundation of China under Grant 62471376 and 62595732, in part by the Natural Science Foundation of Sichuan Province under Grant 2026YFHZ0205, and in part by the IDT Program of Xi'an Jiaotong University under Grant IDT2520.

## Impact Statement

This paper presents work whose goal is to advance the field of machine learning. There are many potential societal consequences of our work, none of which we feel must be specifically highlighted here.

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

## A. Theoretical Analysis of Representational Power

To address the fundamental limitations of prior shape-based attacks, we provide a formal analysis of the representational space offered by our Fourier-based approach.

**Universality of Representation.** Let $\mathcal{C}$ be the space of all continuous, closed, non-self-intersecting curves in $\mathbb{R}^2$. According to the theory of Fourier descriptors (Persoon & Fu, 2007), any curve $\gamma \in \mathcal{C}$ can be approximated by a finite Fourier series $F_K(t)$ to arbitrary precision. Formally, for any $\epsilon > 0$, there exists a minimum frequency $K_\epsilon$ such that $\sup_t \|F(t) - F_{K_\epsilon}(t)\| < \epsilon$. This implies that as $K \to \infty$, the modeling space of our method spans the entire space of physically realizable closed contours (Zhang et al., 2002).

**Comparison with Prior Modeling Spaces.** Unlike grid-based methods (Wei et al., 2024b; 2023d;a)(which are limited by the discrete Nyquist sampling theorem of the grid resolution) or ray-based methods (Chen et al., 2022) (which are strictly restricted to the subset of star-shaped polygons), our Fourier representation defines a hierarchical search space.

For $K = 1$, the space $\mathcal{S}_1$ represents all possible ellipses (including circles), covering basic geometric suppressors.

As $K$ increases, each additional term $c_k$ introduces a new harmonic frequency, expanding the space to $\mathcal{S}_K$. This allows our model to capture complex *concave* and *non-star-shaped* geometries that are mathematically impossible for methods like Def. P (Chen et al., 2022) to represent.

**Differentiable Optimization in Continuous Space.** While top-down spline methods offer similar smoothness, they often lack an analytic relationship between the shape parameters and the resulting image mask. Our Fourier representation, coupled with the winding number theorem (Sec. 3.3), ensures that the mapping $G : \Theta \to M_s$ is not only surjective onto the space of discretized shapes but also differentiable. This allows us to leverage gradient-based optimization to navigate the vast representational space effectively, transforming the search for an optimal shape into a standard continuous optimization problem.

## B. Detailing settings on Comparison Methods

To ensure fairness and reproducibility of comparative experiments, all comparison methods are configured utilizing the optimal hyper-parameters reported in the original paper. Specifically, for UAP, the population number is set to 60 with original patch radius to 30. For Inf.P, the cover rate is set to 0.15, while each adversarial example is optimized for 100 epochs in optimization process. Remaining parameters in reproduced UAP and Inf.P are consistent with the corresponding official open source implementations. For Def.P, to enhance the representation capacity of irregular contours, ray number is set to 100. During patch applying, the size is scaled to 0.6 times target person's height and width. Initialized learning rate is set to 0.002, and each image is optimized for 500 epochs.

## C. Evaluation on Adversarial Augmentation

We further compare the performance of proposed method with other approaches and evaluate the robustness against adversarial augmentation under YOLOv3, Retinanet and Faster R-CNN. Specifically, target detectors are fine-tuned for 1 epoch using generated adversarial example to improve the robustness. Subsequently, adversarial patches are trained and evaluated for attack performance on these augmented detectors. Figure 9(a) compares our methods with other comparison approaches. The highest ASR among all confidence levels after defense demonstrates that our method maintains best robustness-effectiveness trade-off. This advantage stems from the infinity shape representation space, which enables more effective exploration of adversarial shape. Figure 9(b) illustrates the ASR curve before (dash lines) and after (solid lines) adversarial augmentation defense. Results indicate that single-stage detectors like YOLOv3 and RetinaNet exhibit significantly improved resistance to attacks after adversarial augmentation. In contrast, two-stage detector Faster R-CNN shows limited improvement, primarily attributed to the complex architecture preventing effective gradient propagation during adversarial augmentation.

Furthermore, to explore the impact of using different shapes on detector robustness during adversarial augmentation, we fine-tune YOLOv3 detector for 1 epoch using adversarial examples generated with simple geometric shapes (including rectangle, triangle, ellipse, star, polygon, etc.) and optimized Fourier shapes, respectively. We train adversarial patches using augmented detector and evaluate their attack performance. Results are presented in Figure 10. Visualization on left side shows appearances of different shapes on human bodies, while ASR-Conf results on right side show attack performance of

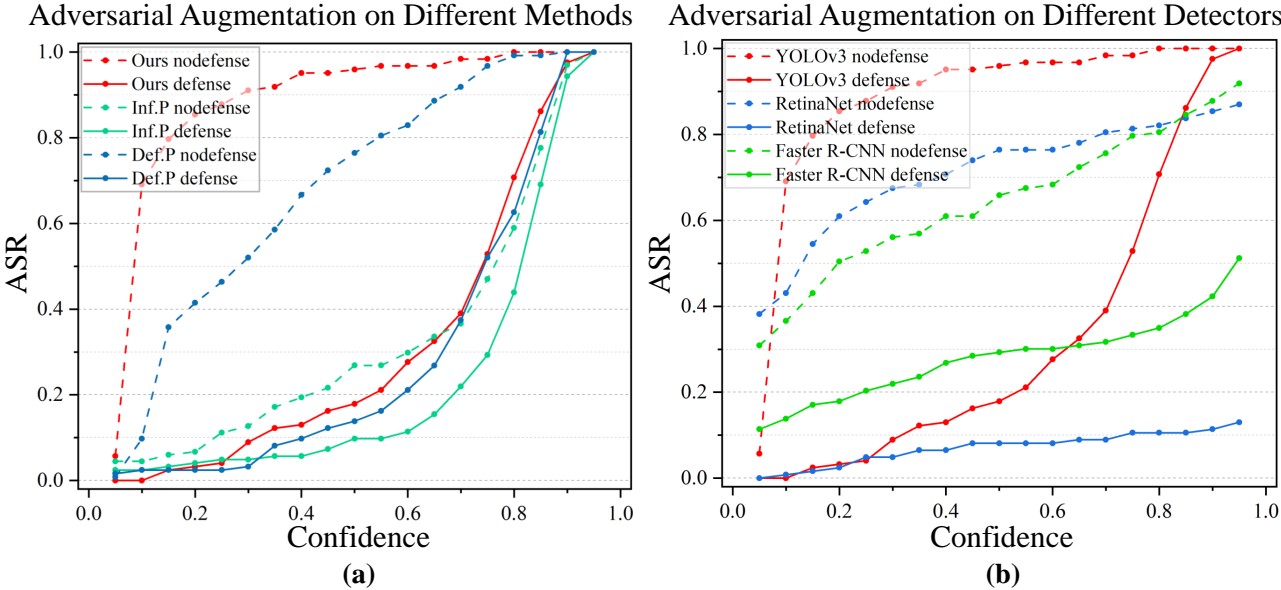

*Figure 9.* Results on adversarial augmentation between different methods, and different detectors. (a) Attack performance of different methods before (dash lines) and after (solid lines) adversarial augmentation on YOLOv3 detector. Our method consistently demonstrates superior robustness under all confidence levels compared to other methods. (b) Attack performance of proposed adversarial shape across different detectors before (dash lines) and after (solid lines) adversarial augmentation, which demonstrates stronger defensive efficacy on single-stage detectors (YOLOv3, RetinaNet) than two-stage detector (Faster R-CNN).

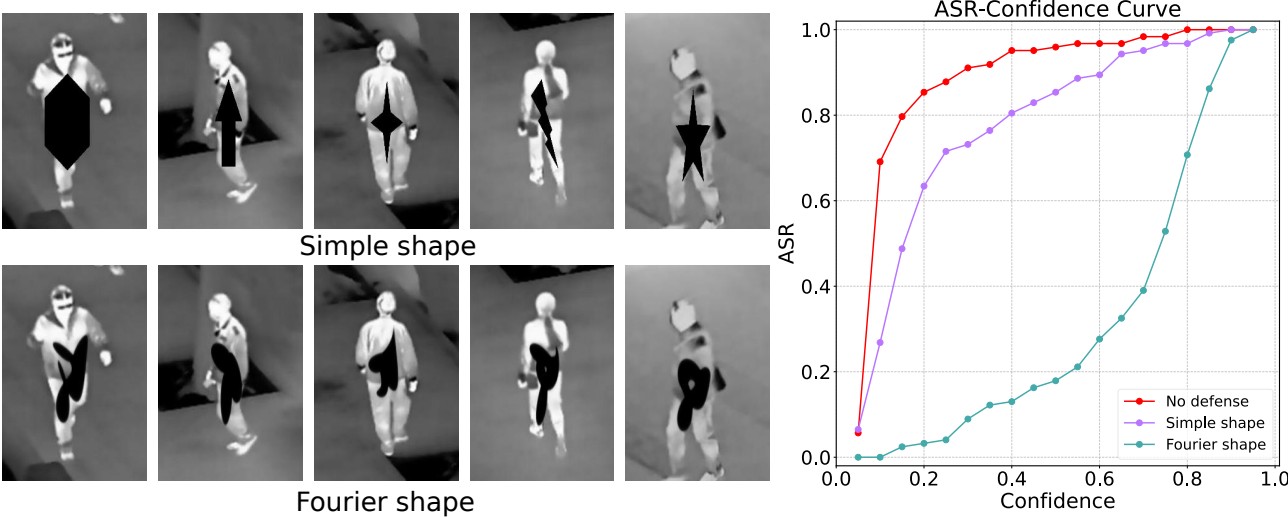

*Figure 10.* Visualization results of patches with different geometric shapes and quantitative ASR-Conf results of proposed adversarial attack against different adversarial augmentation strategies. When Fourier shapes are absent from defense priors, augmentation strategy based on limited regular geometry shapes fails to effectively defense proposed attack method. In contrast, introducing optimized Fourier shapes as defense prior during adversarial augmentation can improve detector's robustness against proposed attack.

our method against different augmentation strategies. The results demonstrate that when detector is augmented solely using known simple geometric shapes as a defense strategy without incorporating optimized Fourier shapes as a defense prior, it fails to effectively defend proposed adversarial attack. In contrast, when optimized Fourier shapes are introduced as defense prior during model augmentation, detector exhibits improved resistance against the attack.

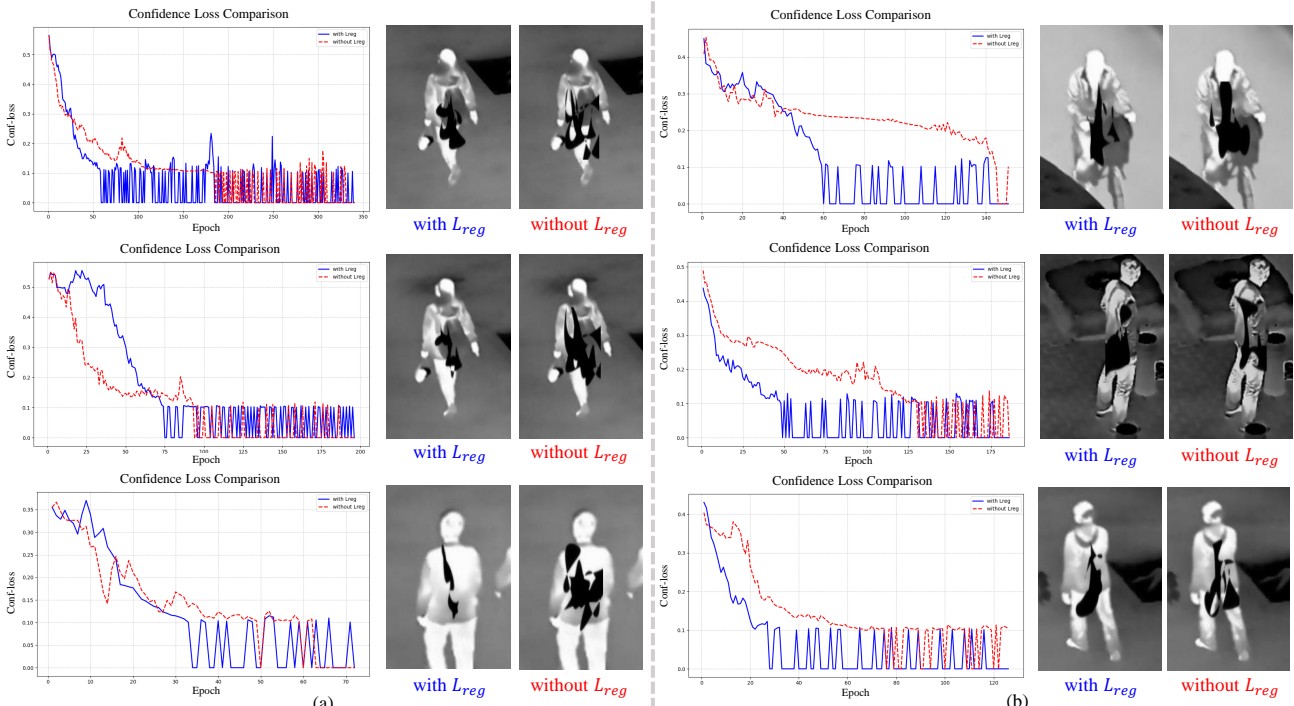

*Figure 11.* Optimization with and without regularization loss $\mathcal{L}_{reg}$. (a) Different shapes with and without $\mathcal{L}_{reg}$. Adversarial shapes optimized with $\mathcal{L}_{reg}$ are significantly simpler than those without $\mathcal{L}_{reg}$. (b) In some cases, $\mathcal{L}_{reg}$ can help adversarial patches converge faster. During the training, we set the confidence loss threshold of 0.1 as the criterion for successful attack and the stopping condition of optimization. When Conf-loss falls below the threshold, it is set to zero. Consequently, in later patch training stage, confidence loss fluctuated slightly between 0 and 0.1, indicating that the adversarial patch has achieved effectiveness and optimization has reached stabilization.

## D. Effectiveness of Regularization Loss

To validate the effectiveness of regularization loss $\mathcal{L}_{reg}$, two experiments are designed. In the first experiment, the initialized Fourier coefficients $\Theta$ is designed to form a complex and self-intersecting Fourier shape. In the second experiment, the initialized $\Theta$ is designed to form a simpler Fourier shape. Experimental results demonstrate that $\mathcal{L}_{reg}$ significantly reduces the complexity of optimized adversarial Fourier shapes, as shown in Figure 11(a), successfully enhancing physical plausibility. Concurrently, $\mathcal{L}_{reg}$ effectively accelerates convergence in attack optimization in some cases, as shown in Figure 11(b).

## E. Implementation Details on Our Proposed Method

**Initialization**: During initialization, dual constraints are applied to $\Theta$. Specifically, low frequency components are optimized to ensure the adversarial shape fully wraps the target spatial region, while high frequency components are constrained to strictly stay within the patch placement boundaries. Through iterative optimization, we derive the Fourier coefficients $\Theta$ that satisfied the constraints. The strategy guarantees that the initialized Fourier shape comprehensively covers the target spatial region while remaining confined within the boundaries of patch placement area, concurrently mitigating complexity induced by excessive curve self-intersection.

**Multi-Detectors**: The training details on different detectors including YOLOv3, YOLOv8, ReteinaNet and Faster R-CNN are presented as follow. Optimization process terminates when either the predicted confidence score on adversarial example remain below a given threshold for 10 consecutive iterations, or the maximum iteration limit is achieved. For YOLOv3 and YOLOv8, the confidence score threshold is set to 0.1, while the maximum iteration limit is set to 1000. For RetinaNet, the optimization strategy remains fundamentally consistent. At the same time, a training schedule is added. When the prediction confidence falls below 0.2, the learning rate decays to 20% of initial value to accelerate convergence. The maximum iteration limit is set to 1500. For Faster R-CNN, to enhance adversarial effectiveness, both confidence score outputs of RPN

and ROI network are applied for optimization. Since the two-stage detector brings higher difficulty on attacking, a more lenient confidence threshold in training schedule is adopted: the learning rate drops to 20% of initial value when predicted confidence of adversarial example drops below 0.7, and maximum iteration limit is set to 2000.

**Augmentation**: To enhance robustness of adversarial attack in physical world, augmented simulations are incorporated in digital domain during optimization, which include applying random translation within a $\pm10$-pixel range, random rotation of $\pm5°$ around the patch center, random scaling between $0.9\times$ and $1.1\times$ of original patch size, and gray-scale disturbance in the range of 0-20 pixel value. By simulating the appearance changes that may be introduced by physical environment, this approach effectively mitigates the representational gap between digital and physical domain.

## F. Adversarial Fourier Shape Pseudo Code

To demonstrate the reproducibility of our proposed method, the pseudo code of generating Fourier curve and calculating differential winding number are shown below. The code can run under environment of Python 3.8 and PyTorch 2.4.1. Note that the code is in the form of pseudo code, which need more adjustment before actually using.

```python
import torch
# Calculating real and imaginary parts of Fourier function
def make_fourier_curve(c):
    N = int(len(c) / 2)
    index = torch.arange(-N, N + 1)

    def X_func(t):
        exponents = torch.tensor(index)
        t_expanded = t.view(-1, 1)
        exp_terms = torch.exp(1j * t_expanded * exponents)
        f_t = torch.sum(c * exp_terms, dim=1)
        return torch.real(f_t)

    def Y_func(t):
        exponents = torch.tensor(index)
        t_expanded = t.view(-1, 1)
        exp_terms = torch.exp(1j * t_expanded * exponents)
        f_t = torch.sum(c * exp_terms, dim=1)
        return torch.imag(f_t)

    return X_func, Y_func

# Calculating winding number of Fourier shape
def differentiable_winding_number(x, y, t, X_func, Y_func)
    t = t.requires_grad_(True)
    X, Y = X_func(t), Y_func(t)

    dX_dt = torch.autograd.grad(X, t,
                                torch.ones_like(X),
                                create_graph=True,
                                retain_graph=True)[0]
    dY_dt = torch.autograd.grad(Y, t,
                                torch.ones_like(Y),
                                create_graph=True,
                                retain_graph=True)[0]

    X, Y = X.unsqueeze(-1), Y.unsqueeze(-1)
    dX_dt = dX_dt.unsqueeze(-1)
    dY_dt = dY_dt.unsqueeze(-1)

    # Discrete differential winding number
    numerator = (X - x) * dY_dt - (Y - y) * dX_dt
    denominator = (X - x) ** 2 + (Y - y) ** 2
    integrand = numerator / denominator
    G = torch.trapz(integrand, t, dim=0) / (2 * np.pi)
    return G
```

```
48  # Example for generating differentiable Fourier mask
49  N = 6
50  c = init(N)
51  t = torch.linspace(0, 2 * np.pi, 1000)
52  X_func, Y_func = make_fourier_curve(c)
53  x_grid = torch.linspace(-0.5, 0.5, 200)
54  y_grid = torch.linspace(-0.5, 0.5, 200)
55  X_mesh, Y_mesh = torch.meshgrid(x_grid, y_grid, indexing='xy')
56  x_flat, y_flat = X_mesh.reshape(-1), Y_mesh.reshape(-1)
57  G_values = differentiable_winding_number(x_flat, y_flat, t, X_func, Y_func)
58  G_values_abs = torch.abs(G_values)
59  mask = torch.clamp(G_values_abs, min=0.0, max=1.0)
```

## G. Video Demo

Video demo of our proposed physical adversarial attack is available in the open-source repository. The video shows a situation that two individuals walking from far to near. One of them carries an adversarial patch, while the other maintains an unmodified state as control. The video reveals that the individual with adversarial patch exhibits significantly lower detection probability compared to the unmodified individual, which demonstrates a successful attack.

