# OpenReview forum: "Unleashing the Representational Power of Fourier Shapes for Attacking Infrared Object Detection"
_ICML.cc/2026/Conference — ICML 2026 regular_

### Official Review · Reviewer_XDTG · 2026-03-05

**Soundness:** 3
**Presentation:** 3
**Significance:** 3
**Originality:** 2
**Overall Recommendation:** 4
**Confidence:** 3

**Summary:**

This paper proposes a differentiable framework for generating physical adversarial attacks against infrared pedestrian detectors. The approach represents the boundary of a heat-blocking patch using Fourier series coefficients, enabling gradient-based optimisation over expressive closed shapes. Theoretical justification on the universality of Fourier representations is provided, and extensive experiments on the LLVIP infrared dataset show stronger, more stable attacks than prior shape-based methods, targeting YOLOv3 and other detectors.

**Compliance With Llm Reviewing Policy:**

Affirmed.

**Final Justification:**

The rebuttal addressed some of my questions/concerns and therefore I keep my original score.

**Key Questions For Authors:**

1. How sensitive is performance to K and regularisation parameter? Are there diminishing returns or instability?
2. How does the attack perform on a broader, more diverse set of images?
3. What is the average optimisation time compared to prior methods?
4. Does the Fourier parameterisation improve cross-detector transfer?
5. Could simple geometric priors or contour regularisation mitigate the attack?

**Limitations:**

The impact statement is minimal and does not meaningfully discuss potential misuse. This work explicitly develops physical attacks against surveillance systems operating in infrared domains, which are widely used in security and autonomous driving. Even a short reflection on safeguards or defensive insights derived from the work would improve transparency.

**Strengths And Weaknesses:**

Strengths:
1. Fourier descriptors for closed contours are well-motivated and mathematically grounded. The differentiable rasterisation step is coherent, and the method supports expressive shape optimisation.
2. Evaluations include ASR–confidence curves, AP drop, and robustness tests under augmentation, compression, and blur. Comparisons with prior work are fair, and experiments span multiple detectors.
3. The paper is clear and structured, with helpful figures illustrating shape parameterisations and comparisons between modelling approaches.
4. This paper addresses an underexplored area of physical adversarial attacks in infrared, shifting focus from texture-based to geometry-based attacks. Adapting Fourier representations to this context is novel, even if the underlying mathematical tool is well-known.

Weaknesses:
1. The paper focuses on representational capacity but lacks deeper discussion of optimisation dynamics, convergence, and sensitivity to the number of Fourier modes (K) or regularisation.
2. Experiments are limited to a curated subset of LLVIP with near-perfect AP, leaving open questions about performance on more diverse or lower-confidence detections.
3. Some theoretical sections read more like justification than actionable contributions. Hyperparameter sensitivity, runtime, and optimisation cost could be reported more clearly.
4. Contribution is incremental and mostly relevant to the adversarial ML subfield. Ethical implications and dual-use considerations are minimally discussed.
5. Cross-detector transferability are not evaluated.

---

> ### Author Rebuttal · Authors · 2026-03-30
>
> ## **Q1: How sensitive is performance to K and regularisation parameter?**
>
> **A1**: Fig. 6 in the main paper already reports the ablation on K. ASR generally increases with K, but with diminishing returns; we use K=6 as a practical balance. The effect of the regularization term is discussed in Supp. D: without high-frequency suppression, the Fourier contour often self-intersects severely, leading to unstable optimization and poor convergence.
>
> ## **Q2: How does the attack perform on a broader, more diverse set of images?**
>
> **A2**: We further evaluate Fourier shape on the DroneVehicle dataset, a UAV-view vehicle dataset containing both RGB and infrared images. We optimize the shape on the 1,469 infrared images in the validation set and evaluate the attack on 8,980 test images. The attack targets all vehicle categories in DroneVehicle, including car, truck, bus, van, and freight car.
>
> | Detector | Avg. area | ASR@0.5 |
> |:---:|:---:|:---:|
> | YOLOv3 | 0.2443 | 0.8118 |
> | YOLOv5 | 0.2667 | 0.8540 |
> | YOLOv8 | 0.3044 | 0.8616 |
>
> Here, area denotes the average percentage of the target bounding box covered by the patch. These results show that, in infrared imagery with strong contour cues, Fourier shape remains highly effective, further supporting the effectiveness of our method across different datasets and attack targets.
>
> ## **Q3: What is the average optimisation time compared to prior methods?**
>
> **A3**: A detailed comparison of the optimization settings and average runtime across different methods is provided in rebuttal Wng3@A2.
>
> ## **Q4: Does the Fourier parameterisation improve cross-detector transfer?**
>
> **A4**: We further evaluate the transfer attack performance of Fourier shape and report ASR@0.5 below:
>
> | Method | Target \ Victim | YOLOv3 | YOLOv8 | RetinaNet | Faster R-CNN |
> |:---:|:---:|:---:|:---:|:---:|:---:|
> | Ours | YOLOv3 | 0.9593 | 0.0488 | 0.1870 | 0.3252 |
> | Ours | YOLOv8 | 0.0163 | 0.7326 | 0.2764 | 0.4228 |
> | Ours | RetinaNet | 0.1057 | 0.4959 | 0.7642 | 0.6667 |
> | Ours | Faster R-CNN | 0.1057 | 0.3496 | 0.3252 | 0.6585 |
> |  |  |  |  |  |  |
> | Def.P | YOLOv3 | 0.7642 | 0.0244 | 0.0650 | 0.0732 |
> | Def.P | YOLOv8 | 0.0081 | 0.4390 | 0.0407 | 0.0569 |
> | Def.P | RetinaNet | 0.0000 | 0.0081 | 0.0813 | 0.1545 |
> | Def.P | Faster R-CNN | 0.0163 | 0.0244 | 0.0813 | 0.1057 |
>
> All shape patches are trained on the target model and tested on the victim model. Our method exhibits clearly stronger cross-model transferability compared to the previous best-performing Def.P.
>
> ## **Q5: Could simple geometric priors or contour regularisation mitigate the attack?**
>
> **A5**: Supplementary Sec. C already discusses defenses that augment the training set with adversarial shapes and simple geometric shapes. Overall, simple shapes do not effectively mitigate the Fourier-based attack, whereas fine-tuning the detector with optimized adversarial shapes improves adversarial robustness. More defense results are provided in rebuttal **ceQi@A3**.
>
> ## **Weaknesses: Ethical implications, misuse risks, and societal consequences.**
>
> **A6**: We agree and will expand the ethical discussion to explicitly address misuse risks and scope boundaries. Our attack could be misused to disrupt the perception results of surveillance, access control, traffic monitoring, and safety systems by reducing detector recall, thereby creating potential safety and security risks.
>
> At the same time, its practical deployment is constrained: the attacker typically needs physical access to the target for patch placement, the effect is valid only under a limited range of viewing conditions, and in the white-box setting it requires knowledge of the target detector’s architecture and parameters, which is often difficult in real deployments. While we observe some black-box transferability, it is still limited and leaves substantial room for improvement.
>
> Beyond attack performance, the scientific value of this work is to reveal the strong sensitivity of current infrared deep detectors to shape attributes. We believe this contributes to understanding model behavior and interpretability, and provides useful evidence for developing more robust detectors and stronger adversarial defenses.

---

> > ### Author Rebuttal · Reviewer_XDTG · 2026-04-01
> >
> > I would keep my original score of 4.

---

### Official Review · Reviewer_Wng3 · 2026-03-10

**Soundness:** 3
**Presentation:** 2
**Significance:** 2
**Originality:** 3
**Overall Recommendation:** 3
**Confidence:** 3

**Summary:**

This paper investigates physical adversarial attacks against infrared target detectors, where the attack signal is primarily carried by the geometry of thermally blocking blocks rather than RGB textures. The authors aim to address a issue: existing shape-based infrared attack methods struggle to balance expressive shape parameterization with efficient optimization. To resolve this, The authors utilize compact Fourier series representations of the block contours and map these contours to differentiable pixel masks using the winding number theorem, achieving gradient-based end-to-end optimization. This research evaluates whether a more expressive and differentiable shape model can significantly improve the performance of physical infrared attacks. Experiments were conducted on the LLVIP dataset with YOLOv3 as the primary target, and the evaluation was extended to RetinaNet, Faster R-CNN, and YOLOv8, covering ablation experiments with different Fourier orders and block scales. Results for both digital and physical attacks are reported.

**Compliance With Llm Reviewing Policy:**

Affirmed.

**Final Justification:**

I hope the authors can verify the effectiveness and generalization of infrared attacks in more  tasks, or explain the principle of infrared failure from a theoretical perspective.

**Key Questions For Authors:**

1.	The paper argues that ASR-confidence curves are more informative than single-threshold ASR. Could the authors also report threshold-integrated summary statistics to make quantitative comparisons across methods easier?
2.	How sensitive is the performance to the exact optimization budget used for the baselines? In particular, are all methods matched in terms of iteration count, area budget, and detector pre-/post-processing settings?
3.	In the physical attack experiments, how robust is the patch under different ambient temperatures, weather conditions, body motion speeds, and non-frontal body orientations beyond the tested range?
4.	The appendix claims that the contour becomes nearly universal in expressiveness as K increases. In practice, what range of K remains physically fabricable and stable under real-world perturbations, and how does this trade off against attack performance?
5.	Since this is a physical attack paper, could the authors expand the discussion of defense implications and misuse risks, rather than addressing societal consequences only briefly?

**Limitations:**

The main limitations are the still limited scale of the physical evaluation, the incomplete discussion of risk and broader impact, and the fact that the strongest empirical evidence is tied more to ASR-threshold behavior than to consistently large gains on conventional metrics such as AP drop. In addition, the current submission contains presentation artifacts that should be fixed before any publication-quality release.

**Strengths And Weaknesses:**

Strengths
1.	Clear technical motivation and well-matched parameterization.
This paper points out a significant bottleneck in previous infrared attack methods: mesh-based methods, while differentiable, are relatively coarse, while spline and ray-based methods, although geometrically superior, have weaker optimization performance. The proposed Fourier contour representation is a reasonable approach that balances expressive power and optimization performance.
2.	Technically coherent end-to-end formulation.
This method combines attack loss, area control, and shape regularization. Overall, the framework is conceptually clear and easy to understand. The appendix also provides a reasonable representational argument explaining why Fourier shapes can cover a richer family of contours than star-shaped or coarse-mesh methods.
3.	Broader evaluation scope.
Beyond the main experiments on YOLOv3, the paper also examines whether the proposed method generalizes to other detectors, including RetinaNet, Faster R-CNN, and YOLOv8. It further reports ablations on the number of Fourier terms and the scale ratio, and extends the evaluation to real-world physical tests under varying distances, poses, viewing angles, and human subjects. Taken together, these additional experiments make the empirical evidence more convincing than results based only on standard numerical benchmarks.
4.	The critique of single-threshold ASR is valuable.
Another strength of the paper is its discussion of the limitations of reporting ASR under only a single confidence threshold. The authors suggest using an ASR-confidence curve for evaluation. I find this to be a meaningful methodological contribution, especially because it offers a more informative and transparent way to assess attack performance.
Weaknesses
1.	The gains are not always as strong as the paper suggests.
Although the ASR-confidence curves are favorable, the AP drop on YOLO models is still relatively limited. Part of the paper’s argument also depends on shifting attention away from conventional summary metrics. This does not invalidate the method, but it does weaken the claim that it is strikingly superior.
2.	The physical evaluation is still somewhat limited.
The physical experiments are promising, but the current setup is still fairly small in scale, with only one material and a limited range of angles. Performance also drops noticeably at closer distances (15m–25m). The evaluation would be stronger with more systematic tests across environments, temperatures, motion conditions, and detector thresholds.
3.	The fairness of the baseline comparison needs clearer discussion.
The appendix states that the reproduced baselines use the optimal hyper-parameters reported in the original papers, but the comparison would be more convincing with clearer matching of patch area budgets, optimization steps, and detector-side settings. Since the main claim concerns the representation-optimization trade-off, fairness here is especially important.

---

> ### Author Rebuttal · Authors · 2026-03-30
>
> ## **Q1: A threshold-integrated metric to facilitate direct quantitative comparison across methods.**
>
> **A1**: To make quantitative comparison easier, we report ASR at several representative confidence thresholds on YOLOv3, together with the mean ASR over thresholds from 0.1 to 0.9. This averaged metric summarizes attack performance across detector operating points, while lower-threshold ASR corresponds to a stricter evaluation for concealment attacks, since the target must remain undetected even when the detector accepts low-confidence predictions.
>
> | Method | ASR@0.3 | ASR@0.5 | ASR@0.7 | Avg. ASR@0.1:0.9 |
> |:---:|:---:|:---:|:---:|:---:|
> | UAP | 0.0075 | 0.0373 | 0.7090 | 0.3151 |
> | Inf.P | 0.1269 | 0.2687 | 0.3657 | 0.3251 |
> | Def.P | 0.5203 | 0.7642 | 0.9187 | 0.6893 |
> | Ours | 0.9106 | 0.9593 | 0.9837 | 0.9241 |
>
> Under both the per-threshold metrics and the threshold-integrated metric, our method performs best, with Avg. ASR@0.1:0.9 = 0.9241, substantially exceeding the baselines.
>
> ## **Q2: How sensitive is the performance to the exact optimization budget used for the baselines?**
>
> **A2**: The patches are optimized offline and then physically fabricated, so real-time update is not required in deployment. Since the methods use different shape parameterizations, optimizers, and stopping rules, matching them by a single iteration count is not very meaningful. Instead, we report the actual optimization cost, method-specific budget, and the realized average patch area below. All methods are optimized on the same 122 images using the same detector pre-/post-processing pipeline.
>
> | Method | Total time on 122 images (s) | Avg. time / image (s) | Optimization budget per image | Simplified stopping rule | Avg. patch area / bbox |
> |:---:|:---|:---|:---|:---|:---:|
> | Def.P | 1081.98 | 8.9 | Avg. 379 iterations (max 500) | Stop early if confidence ≤ 0.1; otherwise stop at 500 iterations. | 0.0700 |
> | Inf.P | 1375.29 | 11.3 | Up to 5 rounds × 100 iterations | Move to the next image once confidence ≤ 0.5; otherwise keep the result after round 5. | 0.1441 |
> | UAP | 28149.26 | 230.7 | Up to 200 generations (population size 60) | Stop once the target fitness is reached; otherwise stop at the maximum generations. | 0.1791 |
> | Ours | 1647.58 | 13.5 | Avg. 301 iterations (max 500) | Stop early if confidence ≤ 0.1; otherwise stop at 500 iterations. | 0.1167 |
>
> These numbers show that our method is not benefiting from a disproportionately large optimization budget. In particular, our patch is not the largest one on average. UAP is much slower because it uses a black-box evolutionary search, which is mainly CPU-driven. Def.P and Inf.P converge faster because they use simpler shape parameterizations. Our method adopts a more expressive parameterization, which may require somewhat more time to identify suitable Fourier coefficients, but the overhead remains moderate and practical for offline optimization. All runtime measurements were obtained on an Intel(R) Xeon(R) Silver 4316 CPU @ 2.30GHz and one NVIDIA GeForce RTX 3090.
>
> ## **Q3: How robust is the patch under different physical attack experiments?**
>
> **A3**: We have evaluated the attack under different distances, viewing angles, poses, and individuals. In addition, we further tested the attack under stronger physical variability, including object motion, camera shake/motion, clothing changes, rear-view placement (placing the patch on the back and imaging from behind), and environment temperature variation. **Please refer to rebuttal KSdB@A2 for detailed results.**
>
> ## **Q4: What practical range of K is physically deployable, and how does it trade off with attack performance?**
>
> **A4**: As the Fourier order K increases, the contour can represent finer geometric details and thus encode richer adversarial semantics. However, in practice, a larger K is not always better. Higher K tends to introduce more high-frequency boundary details, which can: 1) make optimization harder to converge, and 2) make the shape harder to fabricate.
>
> This is also consistent with our ablation in Fig. 6: when K increases from 6 to 8, the ASR drops slightly, likely because the optimization becomes more difficult; with further increases, the ASR improves overall, but the gain shows diminishing returns. Therefore, in this paper we empirically set K=6, which provides a good balance between attack effectiveness and physical manufacturability.
>
> ## **Q5: Defense implications and misuse risks.**
>
> **A5**: We evaluate different defenses through quantitative experiments, which can refer to rebuttal **ceQi@A3**. Results show that introduction of Fourier shape as a priori information in adversarial augmentation can serve as an effective defense method in practical.
>
> Please refer to rebuttal **XDTG@A6** for misuse risks.

---

> > ### Author Rebuttal · Reviewer_Wng3 · 2026-04-03
> >
> > Thank you for the rebuttal. Q1 is satisfactorily addressed: the added multi-threshold ASR and Avg. ASR@0.1:0.9 make cross-method comparison easier. Q2 is partially addressed: the added table clarifies runtime, average iterations, and patch area, but the baselines still use different stopping rules and  optimization schemes, so the fairness concern is not fully removed. Q4 is also reasonably clarified, and the choice of K=6 is now better motivated by the trade-off between expressiveness, optimization difficulty, and fabricability.  I am also concerned about its generalization performance in the real world.

---

> > > ### Author Response · Authors · 2026-04-05
> > >
> > > ## **Q1: The added table clarifies runtime, average iterations, and patch area, but the baselines still use different stopping rules and optimization schemes, so the fairness concern is not fully removed.**
> > >
> > > **A1**: We apologize for not making this clearer in the previous round. We agree that the stopping rules and optimization schemes are not identical across methods. However, **the more important fairness criterion is the maximum optimization budget, rather than whether each method uses the same early-stopping condition. For the three white-box optimization-based methods (Def.P, Inf.P, and our Fourier shape), the maximum number of iterations is the same: 500.**
> > >
> > > There are two reasons for this evaluation setting. **First, these attacks use different shape parameterizations and therefore require method-specific optimizers and hyperparameters.** To avoid underestimating the baselines, we follow the settings provided in their official code/repositories as closely as possible. Enforcing a unified optimizer would not necessarily be fairer, since it could substantially hurt methods that were designed around a particular optimization strategy. **Second, the early-stopping rule is used only to save computation, not to improve attack success unfairly**: if an input satisfies the stopping condition early, it is already a successful attack instance; inputs that do not satisfy it continue optimizing until the shared 500-iteration cap, and only those may remain failures.
> > >
> > > UAP is different because it relies on black-box search rather than white-box gradient optimization. As a result, its optimization cost is much higher (typically around 20× longer than the white-box methods in our experiments) and we did not observe meaningful gains from further increasing its search budget.
> > >
> > > We will revise the paper to make this evaluation setting explicit, so that the fairness of the comparison is clearer.
> > >
> > > ## **Q2: Concerned about the generalization performance in the real world.**
> > >
> > > **A2**: We understand the concern regarding real-world generalization. In the paper, we already reported quantitative physical-world results across different distances, and the originally submitted supplementary material includes a demo video of the attack in the physical setting. In the first-round rebuttal, we further measured ASR under a broader set of conditions, including daytime (person standing still, camera shake, subject/camera moving, different clothing, rear-view placement) and nighttime; these results are provided in our response to KSdB@A2.
> > >
> > > **In addition, we further aligned the experimental setting and evaluated the physical-world performance of different attack methods under the same setup.** Specifically, we fixed the distance between the target and the infrared camera to 20m. For each patch, we recorded 30–40s of video, uniformly sampled around 100 frames, and fed them to the YOLOv3 detector to compute the average ASR.
> > >
> > > | Method | ASR@0.3 | ASR@0.5 | ASR@0.7 |
> > > |---|---:|---:|---:|
> > > | Def.P | 0.0000 | 0.0202 | 0.1111 |
> > > | UAP | 0.0000 | 0.0000 | 0.5691 |
> > > | Inf.P | 0.1808 | 0.5212 | 0.7127 |
> > > | Ours | 0.4607 | 0.6696 | 0.9043 |
> > >
> > > Overall, the physical-world performance is consistent with the digital-domain trend in Fig. 4, and our method shows a clear advantage. Notably, Def.P almost fails in the physical setting. We believe this is because its star-polygon parameterization concentrates adversarial semantics on jagged high-frequency edges, which are easily degraded by fabrication and cutting inaccuracies, causing the attack to deteriorate after physical realization. We will include the corresponding physical attack videos for different methods in the camera-ready supplementary material.
> > >
> > > Regarding the visual results in Fig. 7, **we also provide quantitative evaluations under two dynamic physical scenarios**: continuous viewpoint change with the target angle varying from $-15^\circ$ to $15^\circ$, and pose variation where the target changes from standing to crouching and then back to standing.
> > >
> > > | Condition | ASR@0.3 | ASR@0.5 | ASR@0.7 |
> > > |---|---:|---:|---:|
> > > | Angle | 0.3648 | 0.6351 | 0.7838 |
> > > | Pose | 0.2641 | 0.3679 | 0.5188 |
> > >
> > > These additional results further demonstrate the effectiveness and robustness of our attack in realistic physical scenarios. We hope these results address your concern.

---

### Official Review · Reviewer_KSdB · 2026-03-12

**Soundness:** 3
**Presentation:** 3
**Significance:** 4
**Originality:** 4
**Overall Recommendation:** 5
**Confidence:** 4

**Summary:**

This paper studies physical adversarial attacks on infrared object detection, which propose a differentiable framework that represents patch boundaries with learnable Fourier coefficients and maps them to pixel-space masks via an analytic rendering step, enabling efficient gradient-based optimization. The method is evaluated in both digital and physical settings and compared to prior shape-based attacks, with strong reported attack success and robustness across distance/pose/angle conditions.

**Compliance With Llm Reviewing Policy:**

Affirmed.

**Final Justification:**

During the rebuttal the authors have addressed my main cerins, therefore I am raising the score.

**Key Questions For Authors:**

1. How robust is the learned Fourier shape across unseen infrared sensors and calibration pipelines?

2. Can you report performance under stronger physical variability (weather, motion blur, temperature drift, clothing/material changes)?

3. What is the optimization/runtime cost and fabrication overhead per patch for practical deployment?

**Limitations:**

No. The manuscript should more explicitly discuss limitations and potential negative societal impact, including misuse risk, deployment constraints, and responsible disclosure considerations.

**Strengths And Weaknesses:**

**Strengths**
- Clear problem formulation for infrared-specific attack constraints and a well-motivated representation choice.
- Fourier shape parameterization is elegant: compact, expressive, and optimization-friendly.
- End-to-end differentiable pipeline is technically sound and practical for attack generation.
- Includes physical experiments, which substantially strengthen contribution credibility.
- Evaluation appears broad and includes multiple detectors and comparisons to prior attack methods.

**Weaknesses**
- Threat model assumptions (attacker placement and patch deployment constraints) could be formalized more explicitly.
- Defense evaluation appears limited; stronger adaptive defense benchmarks would improve practical conclusions.
- Real-world robustness under broader environmental factors (weather, sensor model variation, temporal dynamics) could be expanded.
- The impact/limitations discussion is minimal and should better acknowledge misuse risks and scope boundaries.

---

> ### Author Rebuttal · Authors · 2026-03-30
>
> ## **Q1: How robust is the learned Fourier shape across unseen infrared sensors and calibration pipelines?**
>
> **A1**: The attack is tied more to the detection model than to the sensor alone. In practice, different infrared sensors/calibration pipelines are usually paired with different detection models, so the standard and more meaningful test is cross-model transfer. If a detector can infer reliably across multiple sensors, a patch optimized for that detector is expected to remain effective. We therefore report transfer ASR@0.5 below:
>
> | Target \ Victim |YOLOv3|YOLOv8|RetinaNet|Faster R-CNN|
> |:---:|:---:|:---:|:---:|:---:|
> |YOLOv3|0.9593|0.0488|0.1870|0.3252|
> |YOLOv8|0.0163|0.7326|0.2764|0.4228|
> |RetinaNet|0.1057|0.4959|0.7642|0.6667|
> |Faster R-CNN|0.1057|0.3496|0.3252|0.6585|
>
> All shape patches are trained on the target model and tested on the victim model. Our method exhibits a certain degree of cross-model transferability.
>
> ## **Q2 & Weaknesses 3: Attack performance under stronger physical variability.**
>
> **A2**: As already discussed in the main text, we have evaluated the attack under different distances, viewing angles, poses, and individuals. In addition, we further tested the attack under stronger physical variability, including object motion, camera shake/motion, clothing changes, rear-view placement (placing the patch on the back and imaging from behind), and environment temperature variation. For each setting, we collected approximately 100 frames containing the shape patch from 20m and computed the mean ASR (on YOLOv3) over these frames at different detector confidence thresholds. The results are shown below.
>
> | Environment | Setting | ASR@0.3 | ASR@0.5 | ASR@0.7 |
> |---|---|:---:|:---:|:---:|
> | Daytime (18°C) | Person standing still | 0.1600 | 0.3100 | 0.9941 |
> | | Camera shake | 0.1121 | 0.4766 | 0.7757 |
> | | Subject moving | 0.1157 | 0.7355 | 0.9835 |
> |  | Camera moving | 0.2424 | 0.4393 | 0.7954 |
> |  | Different clothing | 0.2474 | 0.9794 | 1.0000 |
> |  | Rear-view placement | 0.4800 | 1.0000 | 1.0000 |
> | Nighttime (10°C) | Person standing still | 0.2308 | 0.9615 | 1.0000 |
>
> Overall, the attack remains fairly robust under these physical variations. Since the adversarial signal is mainly carried by the shape-covered patch region, it is relatively insensitive to moderate changes in temperature, motion, and clothing conditions, and therefore maintains good physical attack effectiveness.
>
> ## **Q3: What is the optimization/runtime cost and fabrication overhead per patch for practical deployment?**
>
> **A3**: In practical deployment, the adversarial patch is optimized offline and then physically fabricated, so real-time optimization is not required during attack execution. Nevertheless, for completeness, we measured the optimization cost of different methods when training input-specific shape patches on 122 images. The results are summarized below.
>
> | Method | Total time on 122 images (s) | Avg. time per image (s) | Avg. iterations / search steps  per image | Simplified stopping rule |
> |---|:---|:---|:---:|:---|
> | Def.P | 1081.98 | 8.9 | 379 | Stop early once confidence ≤ 0.1; otherwise stop at 500 iterations. |
> | Inf. P | 1375.29 | 11.3 | Up to 5 rounds × 100 iterations | Move to the next image once confidence ≤ 0.5; otherwise keep the result after 5 rounds. |
> | UAP | 28149.26 | 230.7 | Up to 200 generations | Stop once the target fitness threshold is reached; otherwise stop at the maximum number of generations. |
> | Ours | 1647.58 | 13.5 | 301 | Stop early once confidence ≤ 0.1; otherwise stop at 500 iterations. |
>
> Because UAP uses a black-box evolutionary search strategy, its optimization is mainly CPU-driven and is therefore substantially slower. In contrast, Def.P and Inf.P adopt relatively simple shape parameterizations and thus converge faster. Our method uses a more expressive parameterized representation, which usually requires slightly more time to identify suitable Fourier coefficients, but the overhead remains moderate. All runtime measurements were obtained on an Intel(R) Xeon(R) Silver 4316 CPU @ 2.30GHz and one NVIDIA GeForce RTX 3090.
>
> ## **Weaknesses 1: Threat model assumptions**
>
> **A4**: In the digital setting, we assume the attacker places a single patch near the center of the target body. The patch is optimized within a predefined support region centered on the target bounding box, whose width and height are both set to 0.6× the target box size. This support region only defines the maximum searchable area; the actual patch is the region enclosed by the optimized shape inside it. In the physical experiments, we keep a similar scale, with the enclosing rectangle of the deployed patch being approximately 60 cm × 40 cm.
>
> ## **Weaknesses 2: More Defense Strategies**
>
> **A5**: Please see rebuttal **ceQi@A3** for more detailed defenses.
>
> ## **Weaknesses 4: The impact/limitations discussion**
>
> **A6**: Please see rebuttal **XDTG@A6** for more impact/limitations discussions.

---

> > ### Author Rebuttal · Reviewer_KSdB · 2026-04-03
> >
> > I have read the rebuttal and my concerns are fully resolved; I will raise my score accordingly.

---

### Official Review · Reviewer_ceQi · 2026-03-13

**Soundness:** 3
**Presentation:** 3
**Significance:** 3
**Originality:** 3
**Overall Recommendation:** 4
**Confidence:** 5

**Summary:**

This paper proposes an adversarial attack method targeting infrared object detection systems. Instead of manipulating pixel values or textures, the authors model adversarial patches as parameterized shapes defined by Fourier series coefficients. These shapes are optimized through a differentiable pipeline that maps shape boundaries to pixel masks using the winding number theorem. The optimized shapes can then be physically fabricated using heat-blocking materials to create adversarial patches that evade infrared detectors. Experimental results demonstrate high attack success rates in both digital and real-world settings.

**Compliance With Llm Reviewing Policy:**

Affirmed.

**Final Justification:**

The rebuttal has addressed my concerns. Therefore, I raised my score to 4.

**Key Questions For Authors:**

1.	In Table 2, why would ASR increase over distance? The author should compare it with no attack.
2.	What’s the rationale behind this method that using Fourier Shapes would be better than baselines?
3.	What defenses might mitigate shape-based adversarial attacks in infrared systems?
4.	What is the setting of the attack? Is it a single-frame attack or a universal one?

**Limitations:**

Yes

**Strengths And Weaknesses:**

Strengths
1.	Introduces an innovative shape-based adversarial representation using Fourier coefficients.
2.	Demonstrates both digital and physical experiments
Weaknesses
1.	Lack of numerical comparison with baselines (related methods mentioned in the paper, no attack, random shape)
2.	Lack of novelty and contribution, since similar settings and attacks have already been proposed and discussed in this area.

---

> ### Author Rebuttal · Authors · 2026-03-30
>
> ## **Q1: In Table 2, why would ASR increases over distance?**
>
> **A1**: ASR is higher at longer distances because fewer fine-grained human thermal details are preserved, so detection relies more on coarse shape cues. The patch therefore contributes a larger fraction of the evidence and more easily dominates the detector response. At shorter distances, richer body details remain visible, making the prediction harder to suppress. In the no-attack baseline, clean persons were consistently detected across the tested distances with confidence (>0.5).
>
> ## **Q2: Why should Fourier shapes outperform the baselines?**
>
> **A2**：The advantage comes from two aspects. First, Fourier series parameterize the shape boundary with a compact but highly expressive representation. As K increases, the search space expands to increasingly fine-grained closed shapes, and in theory can approximate arbitrary contours (discussed in detail in Supplementary-A). Second, our winding-number-based differentiable mapping provides an analytic bridge from shape parameters to the pixel mask, enabling efficient gradient-based optimization instead of inefficient black-box evolutionary search.
>
> ## **Q3: What defenses might mitigate our attack?**
>
> **A3**: Adversarial augmentation is the most effective defense in our tests, i.e., retraining the detector with Fourier shapes can substantially improve robustness; detailed results are provided in Supplementary-C. We further evaluated Gaussian blur, JPEG compression, and two additional defenses added in the rebuttal, Digital Watermarking [1] and Local Gradients Smoothing [2]. The ASR results on YOLOv3 at conf.=0.5 are:
> |Defense| ASR (%) |
> |:---|:---:|
> | No defense | 95.93 |
> | Digital Watermarking | 81.30 |
> | JPEG compression | 80.49 |
> | Gaussian blur | 78.05 |
> | Local Gradients Smoothing | 72.36 |
> | Adversarial Augmentation | 16.26 |
>
> Overall, standard image-processing defenses provide only limited mitigation, because the adversarial information is mainly encoded in the patch shape region and is not easily removed by such transformations. In contrast, adversarial augmentation is much more effective, likely because the detector learns part of the shared shape-induced disturbance patterns during training.
>
> ## **Q4: What is the setting of the attack?**
>
> **A4**: Following prior works [3,4], all digital experiments are conducted as single-frame attacks. For the physical evaluation, the Fourier-shape patch is still optimized on a single frame, then deployed and tested under varying conditions, including different distances, viewing angles, and poses. As shown in Fig. 7 in the submitted manuscript, the patch still exhibits clear generalization ability in the physical world despite being trained in an input-specific manner.
>
> ## **Weaknesses 1: Lack of numerical comparison with baselines.**
>
> **A5**: Fig. 4 already provides the ASR-confidence curves of our method and all baselines, which offer a full quantitative comparison across detection thresholds. For additional clarity, we report ASR at several representative confidence levels below.
> | Method | ASR@0.3 | ASR@0.5 | ASR@0.7 | Avg. ASR@0.1:0.9 |
> |---|:---:|:---:|:---:|:---:|
> | Random shape | 0.0164 | 0.1311 | 0.2213 | 0.1990 |
> | UAP | 0.0075 | 0.0373 | 0.7090 | 0.3151 |
> | Inf.P | 0.1269 | 0.2687 | 0.3657 | 0.3251 |
> | Def.P | 0.5203 | 0.7642 | 0.9187 | 0.6893 |
> | Ours | 0.9106 | 0.9593 | 0.9837 | 0.9241 |
>
> These results are consistent with Fig. 4: our method achieves the strongest attack performance across thresholds, while prior methods are much more sensitive to the confidence setting.
>
> ## **Weaknesses 2: Lack of novelty and contribution.**
>
> **A6**: The novelty and contribution are two-fold. First, we propose a novel infrared adversarial patch representation based on Fourier series parameterization. The novelty is not the use of Fourier series in isolation, but its formulation for infrared physical attacks, where shape, rather than texture, is the primary adversarial carrier. Our method can theoretically represent any arbitrary 2D closed shape, significantly expanding the search space and attack potential compared to existing methods. Second, we leverage Winding Number Theory to establish a differentiable mapping from Fourier contours to shape masks. This pipeline enables end-to-end gradient optimization of physically realizable shapes, the efficacy of which is validated through a comprehensive digital-to-physical evaluation.
>
>
> [1] On visible adversarial perturbations & digital watermarking, in CVPR, 2018.
>
> [2] Local gradients smoothing: Defense against localized adversarial attacks, in WACV, 2019.
>
> [3] Infrared Adversarial Patches with Learnable Shapes and Locations in the Physical World, IJCV, 2024.
>
> [4] Unified Adversarial Patch for Visible-Infrared Cross-modal Attacks in the Physical World, TPAMI, 2023.

---

> > ### Author Rebuttal · Reviewer_ceQi · 2026-04-01
> >
> > Thank the authors for the rebuttal. I like the exploration of the defense methods. Here are more questions for the authors, but they need not be addressed in this rebuttal: Does adversarial augmentation transfer well across different physical attack methods? Is adversarial augmentation enough for defense against these kinds of attacks?
> >
> > My concerns about "Weaknesses 1: Lack of numerical comparison with baselines" remain unaddressed. Since the paper focuses on the physical adversarial attack scenario, a numerical comparison with baselines in the physical world is required, and this should be the paper's most important result. Figure 7 is more of a demonstration than a rigorous analysis, as it can typically be cherry-picked. Table 2 is the only numerical result in the physical world; however, a lack of comparison could lead to a false sense of attack effectiveness, and it is hard to cross-validate in future work.

---

> > > ### Author Response · Authors · 2026-04-05
> > >
> > > ## **Q1. Numerical comparison with baselines in the physical world.**
> > >
> > > **A1**: We agree that a numerical physical-world comparison with baselines is essential. To address this concern, we aligned the physical evaluation protocol across methods as closely as possible. Specifically, we fixed the distance between the target and the infrared camera to 20m. For each patch, we recorded 30–40s of video, uniformly sampled to around 100 frames, and fed them to the YOLOv3 detector to compute the average ASR.
> > >
> > > | Method | ASR@0.3 | ASR@0.5 | ASR@0.7 |
> > > |---|---:|---:|---:|
> > > | Def.P | 0.0000 | 0.0202 | 0.1111 |
> > > | UAP | 0.0000 | 0.0000 | 0.5691 |
> > > | Inf.P | 0.1808 | 0.5212 | 0.7127 |
> > > | Ours | 0.4607 | 0.6696 | 0.9043 |
> > >
> > > Overall, the physical-world performance is consistent with the digital results in Fig. 4 of the paper, and our method shows a clear advantage. Notably, Def.P almost fails in the physical setting. We believe this is because its star-polygon parameterization concentrates adversarial semantics on jagged high-frequency edges, which are easily degraded by fabrication/cutting inaccuracies, causing the attack to collapse after physical realization. We will include the corresponding physical attack videos for different methods in the camera-ready supplementary materials.
> > >
> > > Regarding the visual results in Fig. 7 under different physical conditions, we also provide quantitative evaluations for two dynamic scenarios: continuous viewpoint change with the target angle varying from $-15^\circ$ to $15^\circ$, and pose variation where the target changes from standing to crouching and then back to standing. The results are summarized below.
> > >
> > > | Condition | ASR@0.3 | ASR@0.5 | ASR@0.7 |
> > > |---|---:|---:|---:|
> > > | Angle | 0.3648 | 0.6351 | 0.7838 |
> > > | Pose | 0.2641 | 0.3679 | 0.5188 |
> > >
> > > In addition, in the originally submitted supplementary material, we provided a physical-world demo video in which the target continuously walks from far to near. Together, these quantitative results and videos further demonstrate the effectiveness and robustness of our attack in realistic physical scenarios. We hope these additional results address your concern.
> > >
> > > ## **Q2: Does adversarial augmentation transfer well across different physical attack methods?**
> > >
> > > **A2**: We agree that the transferability of adversarial augmentation across different physical attack methods is an interesting and important question. Due to the limited rebuttal time, we are not yet able to provide a complete cross-attack evaluation. To make the discussion precise, we define such transferability as follows: a detector is trained using adversarial examples generated by one attack method, and is then evaluated against a different attack method.
> > >
> > > Our current view is that **the transfer effect of adversarial augmentation is likely limited, and that adversarial augmentation alone is unlikely to be a sufficient defense against this class of attacks.** The main reason is that **different physical attacks rely on different adversarial semantics**. For example, Def.P mainly concentrates its adversarial effect on high-frequency jagged boundaries, whereas Inf.P encodes adversarial cues not only in the shape itself but also in the placement/location pattern, since its rasterized parameterization provides relatively coarse shape modeling. Because the exploited features differ across attacks, robustness gained from augmentation with one attack may not generalize well to another.
> > >
> > > However, a more promising direction may be to train detectors on an ensemble of adversarial shapes generated by multiple attack methods, so that the model is exposed to a broader range of adversarial semantics. We believe this could further improve physical-world robustness.

---

### Decision · Program_Chairs · 2026-04-30

**Decision:**

Accept (regular)

**Comment:**

Most reviewers were positive after the rebuttal, and the majority acknowledged the core contribution. The rebuttal added results and clarifications that sufficiently support the paper’s main claims.
One reviewer still raises a negative concern, mainly asking for stronger experimental evidence, especially in the physical setting. AC views this as a limitation of evaluation breadth rather than a fundamental flaw in the contribution.
Overall, since the main contribution is supported by the paper and strengthened by the rebuttal, and most reviewers ended up on the positive side, AC believes the paper meets the bar for weak accept.